# On the Transferability of Visually Grounded PCFGs

**Yanpeng Zhao**[ɛ]
[ɛ]ILCC, University of Edinburgh
`yanp.zhao@ed.ac.uk`

**Ivan Titov**[ɛæ]
[æ]ILLC, University of Amsterdam
`ititov@inf.ed.ac.uk`

## Abstract

There has been a significant surge of interest in visually grounded grammar induction in recent times. While a variety of models have been developed for the task and have demonstrated impressive performance, they have not been evaluated on text domains that are different from the training domain, so it is unclear if the improvements brought by visual groundings are transferable. Our study aims to fill this gap and assess the degree of transferability. We start by extending VC~PCFG (short for Visually-grounded Compound PCFG (Zhao and Titov, 2020)) in such a way that it can transfer across text domains. We consider a zero-shot transfer learning setting where a model is trained on the source domain and is directly applied to target domains, without any further training. Our experimental results suggest that: the benefits from using visual groundings transfer to text in a domain similar to the training domain but fail to transfer to remote domains. Further, we conduct data and result analysis; we find that the lexicon overlap between the source domain and the target domain is the most important factor in the transferability of VC~PCFG.

## 1 Introduction

Research in unsupervised grammar induction has long focused on learning grammar models from pure text (Lari and Young, 1990; Klein and Manning, 2002; Jiang et al., 2016; Kim et al., 2019). Observing that human language learning is largely grounded in perceptual experiences, a new trend in the area has been inducing grammars of language with visual groundings (Shi et al., 2019; Zhao and Titov, 2020; Hong et al., 2021; Zhang et al., 2021). These grounded models are typically trained on parallel images and text such as image captioning data. Once trained, they can be directly used to parse text without access to the aligned images. Though these studies have concluded that visual grounding is beneficial, they evaluated their models only on in-domain text; it is unclear if the improvements transfer across domains or, equivalently, if the models acquire a general grammar or only a grammar suitable for a certain domain.

In this work, we bridge the gap by studying the transferability of VC~PCFG, a performant visually grounded PCFG model (Zhao and Titov, 2020). To enable VC~PCFG to transfer across different domains, we extend it by using pre-trained word embeddings and obtain transferrable VC~PCFG (model dubbed TVC~PCFG). This modification allows for directly applying VC~PCFG to a target domain, without requiring training on any data from the target domain.

We learn TVC~PCFG on the MSCOCO image-caption pairs (Lin et al., 2014; Chen et al., 2015) and consider two evaluation setups: (1) proximate-domain transfer evaluates the model on the captions from another image-captioning data set; and (2) remote-domain transfer evaluates the model on sentences from 14 non-caption text domains, ranging from news, biography, and fiction to informal web text. Our experiments show that VC~PCFG can transfer to similar text domains but struggle on text domains that are different from the training domain.

To investigate the factors that influence the transferability of TVC~PCFG, we further conduct an analysis of the source- and target-domain data and the parsing results. We propose to measure the distance between the source domain and the target domain via the overlaps of phrasal labels, grammar rules, and words. Our findings suggest that lexicon overlap plays an important role in the transferability of TVC~PCFG. For example, when tested on Wall Street Journal data (Marcus et al., 1999), we observe that the fewer words from a test sentence are present in the MSCOCO data, the lower the performance tends to be.

## 2 Transferable vc~PCFG

### 2.1 Visually-grounded Compound PCFG

To motivate transfer learning of vc~PCFG, we first reiterate the learning objective of vc~PCFG. Suppose a captioning data set $\mathcal{D} = \{(\mathrm{v}^{(i)}, \boldsymbol{w}^{(i)}) | 1 \leq i \leq N\}$ consists of $N$ pairs of image v and caption $\boldsymbol{w}$, the loss function of vc~PCFG consists of a language modeling loss defined on raw text $-\log p(\boldsymbol{w})$ and a contrastive learning loss defined on image-text pairs $s(\mathrm{v}, \boldsymbol{w})$. Formally,

$$\mathcal{L} = \sum_i -\log p(\boldsymbol{w}^{(i)}) + \alpha \cdot s(\mathrm{v}^{(i)}, \boldsymbol{w}^{(i)}), \quad (1)$$

where the hyperparameter $\alpha$ controls the relative importance of the two loss terms.

The language modeling loss from each sentence $\boldsymbol{w}$ is defined as the negative log probability of the sentence. As with a PCFG $\mathcal{G}$, the sentence probability $p(\boldsymbol{w})$ is formulated as the marginalization of all parse trees $\mathcal{T}_{\mathcal{G}}(\boldsymbol{w})$ that yield $\boldsymbol{w}$: $p(\boldsymbol{w}) = \sum_{t \in \mathcal{T}_{\mathcal{G}}(\boldsymbol{w})} p(t)$. It can be tractably computed using the inside algorithm (Baker, 1979).

The contrastive learning loss $s(\mathrm{v}, \boldsymbol{w})$ is defined as a hinge loss. Intuitively, it is optimized to score higher for an aligned pair $(\mathbf{v}^{(i)}, \boldsymbol{w}^{(i)})$ than for any un-aligned pair (i.e., $(\mathbf{v}^{(i)}, \boldsymbol{w}^{(j)})$ or $(\mathbf{v}^{(j)}, \boldsymbol{w}^{(i)})$ with $i \neq j$) by a positive margin. Since $s(v, \boldsymbol{w})$ is computed with respect to $p_{\mathcal{G}}(t|\boldsymbol{w})$, the conditional probability distribution of the parse trees of the sentence $\boldsymbol{w}$ under the PCFG $\mathcal{G}$, optimizing $s(v, \boldsymbol{w})$ will backpropagate the learning signals derived from image-text pairs to the parser $\mathcal{G}$ (see Zhao and Titov (2020) for technical details).

Essentially, the loss function defined in Equation 1 corresponds to multi-objective learning and can be applied to text alone or to image-text pairs. Specifically, for sentences that are paired with aligned images, both the LM loss and the hinge loss are minimized; for sentences without aligned images, only the LM loss is minimized. By treating images as the labels of the aligned text, this type of multi-objective learning can be seen as semi-supervised learning.

### 2.2 Transfer Learning Model

In this work, we consider a zero-shot transfer learning setting: we directly apply and transfer a pre-trained vc~PCFG to the target domain. This setting is viable because vc~PCFG can be learned solely on text and does not rely on images to parse text at inference time.

To enable vc~PCFG to transfer across domains, we extend it by using pre-trained word embeddings and sharing them between the source domain and the target domain. Following the definition by Zhao and Titov (2020), a vc~PCFG consists of three types of grammar rules:

$$\text{Start Rule:} \quad S \rightarrow A, \quad (2)$$
$$\text{Binary Rule:} \quad A \rightarrow BC, \quad (3)$$
$$\text{Preterminal Rule:} \quad T \rightarrow w. \quad (4)$$

The start rules and the binary rules are domain-independent because they are composed of domain-agnostic grammar symbols (e.g., $S$ and $A$), but the preterminal rules, which generate a word conditioning on a preterminal, are domain-dependent since they rely on the domain-specific vocabulary (i.e., $w$). Thus, the key to transferring vc~PCFG from the source domain to the target domain is to share preterminal rules or, equivalently, a vocabulary between the source and target domains.

Still, sharing the same set of grammar rules between the source and target domains does not guarantee that a learned model transfers to unseen preterminal rules. This is because the target-domain vocabulary is not necessarily subsumed by the source-domain vocabulary. To make it more clear, we first note that vc~PCFG generates rule probabilities conditioning on grammar symbols. Take preterminal rules of the form $T \rightarrow w$,

$$p(T \rightarrow w) \propto g(\mathbf{u}_T, \mathbf{e}_w, \mathbf{z}; \theta), \quad (5)$$

where $g_\theta$ is a neural network, $\mathbf{u}$ and $\mathbf{e}$ indicate preterminal-symbol embeddings and word embeddings, respectively, and $\mathbf{z}$ is a sentence-dependent latent vector. Since we train vc~PCFG only on the source domain, for preterminal rules that contain words outside of the source domain, their rule probabilities and, specifically, the word embeddings that are used to compute the rule probabilities, will never be learned.

To resolve this issue, we use pre-trained word embeddings, namely GloVe (Pennington et al., 2014), and refer to the resulting model as tvc~PCFG. Pre-trained word embeddings have encoded similarities among words, i.e., similar words are generally closer in the learned vector space. We keep pre-trained word embeddings frozen during training; thus, at test time, for words (preterminal rules) that are unseen during training, our tvc~PCFG can exploit similarities in the embedding space to estimate rule probabilities.

| Model | NP | VP | PP | SBAR | ADJP | ADVP | C-F1 | S-F1 |
|---|---|---|---|---|---|---|---|---|
| Left Branching | 33.2 | 0.0 | 0.0 | 4.9 | 0.0 | 0.0 | 15.1 | 15.7 |
| Right Branching | 23.8 | 91.5 | 63.0 | **96.0** | 18.3 | 76.7 | 42.4 | 42.8 |
| Random Trees | $32.8_{\pm0.5}$ | $18.4_{\pm0.4}$ | $24.4_{\pm0.3}$ | $17.7_{\pm1.7}$ | $26.8_{\pm2.6}$ | $20.9_{\pm1.5}$ | $24.2_{\pm0.3}$ | $24.6_{\pm0.2}$ |
| c~PCFG [†] | $43.0_{\pm8.6}$ | **$85.0_{\pm2.6}$** | $78.4_{\pm5.6}$ | **$90.6_{\pm2.1}$** | $36.6_{\pm21}$ | **$87.4_{\pm1.0}$** | $53.6_{\pm4.7}$ | $53.7_{\pm4.6}$ |
| vc~PCFG [†] | $54.9_{\pm14}$ | $83.2_{\pm3.9}$ | **$80.9_{\pm7.9}$** | $89.0_{\pm2.0}$ | $38.8_{\pm25}$ | $86.3_{\pm4.1}$ | $59.3_{\pm8.2}$ | $59.4_{\pm8.3}$ |
| tc~PCFG [*] | $31.8_{\pm13.5}$ | $60.0_{\pm25.5}$ | $54.5_{\pm14.0}$ | $73.0_{\pm18.5}$ | $39.3_{\pm23.0}$ | $59.5_{\pm19.4}$ | $38.7_{\pm2.6}$ | $38.8_{\pm2.6}$ |
| PERM | $26.0_{\pm2.8}$ | $36.5_{\pm12.9}$ | $36.0_{\pm2.3}$ | $32.6_{\pm10.9}$ | $25.0_{\pm1.8}$ | $31.2_{\pm0.7}$ | $27.0_{\pm2.0}$ | $27.5_{\pm2.1}$ |
| tvc~PCFG [*] | **$79.1_{\pm6.0}$** | $67.8_{\pm13.7}$ | $71.4_{\pm8.5}$ | $80.7_{\pm9.2}$ | **$59.1_{\pm17.9}$** | $84.9_{\pm3.0}$ | **$65.7_{\pm2.1}$** | **$66.3_{\pm2.1}$** |
| PERM | $42.0_{\pm2.7}$ | $34.2_{\pm5.2}$ | $41.4_{\pm1.9}$ | $33.2_{\pm4.4}$ | $25.4_{\pm2.3}$ | $35.9_{\pm3.8}$ | $35.2_{\pm1.0}$ | $36.0_{\pm1.1}$ |

Table 1: Parsing performance on MSCOCO. [†] indicates the results from Zhao and Titov (2020) and [*] indicates models with pre-trained GloVe word embeddings.

| Model | NP | VP | PP | SBAR | ADJP | ADVP | C-F1 | S-F1 |
|---|---|---|---|---|---|---|---|---|
| Left Branching | 32.9 | 0.0 | 0.3 | 0.5 | 0.5 | 0.0 | 14.4 | 16.4 |
| Right Branching | 27.9 | **88.0** | 56.0 | **92.5** | 13.3 | 66.9 | 44.3 | 48.0 |
| Random Trees | $30.6_{\pm0.2}$ | $17.4_{\pm0.5}$ | $21.9_{\pm0.6}$ | $15.8_{\pm1.8}$ | $25.5_{\pm2.5}$ | $19.6_{\pm5.1}$ | $22.0_{\pm0.3}$ | $24.2_{\pm0.3}$ |
| c~PCFG [†] | $35.6_{\pm23.4}$ | $64.6_{\pm9.0}$ | $63.3_{\pm25.0}$ | $55.1_{\pm33.0}$ | $10.2_{\pm4.4}$ | $58.6_{\pm36.2}$ | $43.0_{\pm16.6}$ | $45.8_{\pm17.3}$ |
| vc~PCFG [†] | $33.7_{\pm20.7}$ | $61.6_{\pm6.2}$ | $46.8_{\pm26.8}$ | $40.6_{\pm37.8}$ | $12.7_{\pm9.1}$ | $39.2_{\pm39.0}$ | $38.0_{\pm15.4}$ | $40.9_{\pm15.4}$ |
| tc~PCFG [*] | $29.6_{\pm15.5}$ | $58.3_{\pm19.1}$ | $58.0_{\pm12.4}$ | $66.7_{\pm9.1}$ | $38.6_{\pm27.2}$ | $55.8_{\pm15.4}$ | $38.5_{\pm2.1}$ | $40.5_{\pm2.0}$ |
| PERM | $22.3_{\pm3.2}$ | $35.4_{\pm9.9}$ | $32.3_{\pm2.0}$ | $27.6_{\pm4.4}$ | $20.5_{\pm0.8}$ | $28.5_{\pm1.7}$ | $24.4_{\pm1.4}$ | $27.3_{\pm1.6}$ |
| tvc~PCFG [*] | **$76.3_{\pm6.5}$** | $64.8_{\pm11.1}$ | **$72.7_{\pm5.6}$** | $69.1_{\pm3.6}$ | **$55.1_{\pm17.9}$** | **$70.0_{\pm4.6}$** | **$63.0_{\pm2.2}$** | **$66.6_{\pm2.3}$** |
| PERM | $37.4_{\pm2.9}$ | $32.8_{\pm4.6}$ | $37.2_{\pm1.6}$ | $30.0_{\pm2.6}$ | $24.8_{\pm2.0}$ | $34.5_{\pm2.4}$ | $31.5_{\pm1.3}$ | $35.5_{\pm1.4}$ |

Table 2: Parsing performance on Flick. [†] indicates the results obtained by running (v)c~PCFG on Flick and [*] indicates the best models (w/ pre-trained GloVe word embeddings) trained on MSCOCO but evaluated on Flickr.

## 3  Experiments

### 3.1  Data Sets and Evaluation

We use MSCOCO captioning data set (Lin et al., 2014; Chen et al., 2015) as the source domain and conduct *proximate-domain* transfer and *remote-domain* transfer experiments. For proximate-domain transfer, we consider Flickr30k (Flickr; Young et al. (2014)). For remote-domain transfer, we consider Wall Street Journal (WSJ) and Brown portions of the Penn Treebank (Marcus et al., 1999), and the English Web Treebank (Enweb; Bies et al. (2012)). Note that Brown and Enweb consist of 8 and 5 subdomains, respectively, so we will be actually performing remote-domain transfer on 14 text domains (see below for details).

**Flickr** is an image captioning dataset. While the images of Flickr and MSCOCO are all sourced from Flickr, they focus on different aspects,[1] so do their captions. Though the guidelines for collecting

MSCOCO captions (Chen et al., 2015) are inspired by those of Flickr (Hodosh et al., 2013; Young et al., 2014), due to the differences in the instructions, the statistics of the collected captions tend to be different, e.g., Flickr test captions are slightly longer than MSCOCO training captions (i.e., 12.4 vs 10.5 tokens on average). Nevertheless, Flickr is close to MSCOCO and thus we choose Flickr captions for *proximate-domain* transfer. Since Flickr does not contain gold phrase structures of captions, we follow Zhao and Titov (2020) and parse all the captions with Benepar (Kitaev and Klein, 2018).

**WSJ** is a news corpus and the central part of the Penn Treebank resource (Marcus et al., 1999). Sentences in WSJ have been manually annotated with phrase structures. We use WSJ for *remote-domain* transfer; sentences in newswire and image captions are very different as evident, for example, from the divergences in distributions of tokens, syntactic fragments, and sentence lengths (20.4 vs 10.5 tokens on average).

**Brown** is also part of the Penn Treebank resource and consists of manually parsed sentences from

---

[1]Flickr images focus on people and animals that perform some actions (Hodosh et al., 2013; Young et al., 2014; Plummer et al., 2015) while MSCOCO covers more diverse object categories (up to 80) and focuses on multiple-object images (Lin et al., 2014; Chen et al., 2015).

| Model | NP | VP | PP | SBAR | ADJP | ADVP | C-F1 | S-F1 |
|---|---|---|---|---|---|---|---|---|
| Left Branching | 10.4 | 0.5 | 5.0 | 5.3 | 2.5 | 8.0 | 6.0 | 8.7 |
| Right Branching | 24.1 | **71.5** | 42.4 | **68.7** | 27.7 | 38.1 | 36.1 | 39.5 |
| Random Trees | $22.5_{\pm0.3}$ | $12.3_{\pm0.3}$ | $19.0_{\pm0.5}$ | $9.3_{\pm0.6}$ | $24.3_{\pm1.7}$ | $26.9_{\pm1.3}$ | $15.3_{\pm0.1}$ | $18.1_{\pm0.1}$ |
| c~PCFG [†] | **$76.7_{\pm2.0}$** | $40.7_{\pm5.5}$ | **$71.3_{\pm2.1}$** | $53.8_{\pm3.1}$ | **$45.9_{\pm2.8}$** | **$64.2_{\pm2.8}$** | **$53.5_{\pm1.4}$** | **$55.7_{\pm1.3}$** |
| ʟ10c~PCFG [†] | $67.1_{\pm3.8}$ | $31.0_{\pm9.8}$ | $61.3_{\pm2.2}$ | $45.9_{\pm8.2}$ | $36.7_{\pm2.3}$ | $41.3_{\pm6.0}$ | $45.5_{\pm2.4}$ | $48.2_{\pm2.3}$ |
| ᴛc~PCFG [*] | $30.9_{\pm5.5}$ | $23.6_{\pm7.3}$ | $36.4_{\pm9.0}$ | $27.2_{\pm5.3}$ | $24.7_{\pm1.6}$ | $34.2_{\pm4.7}$ | $24.4_{\pm1.7}$ | $28.0_{\pm1.9}$ |
| PERM | $20.7_{\pm0.3}$ | $18.0_{\pm3.3}$ | $21.1_{\pm1.2}$ | $13.8_{\pm2.3}$ | $23.0_{\pm0.3}$ | $26.6_{\pm1.6}$ | $16.4_{\pm0.9}$ | $20.0_{\pm1.1}$ |
| ᴛᴠc~PCFG [*] | $48.6_{\pm3.7}$ | $24.8_{\pm4.1}$ | $39.4_{\pm6.5}$ | $27.2_{\pm1.1}$ | $30.2_{\pm4.6}$ | $40.4_{\pm2.0}$ | $32.0_{\pm1.2}$ | $35.3_{\pm1.3}$ |
| PERM | $23.3_{\pm0.4}$ | $17.6_{\pm1.9}$ | $23.6_{\pm0.3}$ | $13.6_{\pm0.6}$ | $23.5_{\pm1.5}$ | $28.6_{\pm2.6}$ | $17.7_{\pm0.5}$ | $21.3_{\pm0.5}$ |

Table 3: Parsing performance on WSJ. [†] indicates the models that are trained and evaluated on WSJ (Zhao and Titov, 2021). The prefix "ʟ10" indicates that the models are trained on WSJ sentences shorter than 11 tokens but are tested on the full WSJ test set. [*] indicates the best models (w/ pre-trained GloVe word embeddings) trained on MSCOCO but evaluated on WSJ.

8 domains, which cover various genres such as lore, biography, fiction, and humor (Marcus et al., 1999). We divide the sentences in each domain into three parts: around 70% of the sentences for training, 15% for development, and 15% for test. We further merge the training, development, and test subsets across domains and create a mixed-domain Brown (see Table 7 in Appendix). The average length of Brown test sentences is much longer than MSCOCO training captions, i.e., 17.1 vs 10.5 tokens. Since all these subdomains differ in terms of genre from image captions, we use them for *remote-domain* transfer.

**Enweb** is short for English Web Treebank and consists of sentences from 5 domains: weblogs, newsgroups, email, reviews, and question-answers (Bies et al., 2012). Each of these domains contains sentences that have been manually annotated with syntactic structures. We divide sentences in each domain in a similar way as we divide Brown sentences. We also create a mixed-domain Enweb (see Table 8 in Appendix). Enweb test sentences are slightly longer than MSCOCO training captions, i.e., 13.9 vs 10.5 tokens on average. Since they belong to genres different from image captions, we use them for *remote-domain* transfer.

### 3.2 Model Configurations

**Transfer learning models.** We use the same implementations of the text-only parser c~PCFG and the visually-grounded version ᴠc~PCFG as Zhao and Titov (2020) but replace their word embeddings with pre-trained GloVe embeddings (models are dubbed ᴛ(ᴠ)c~PCFG). We follow the setups in Zhao and Titov (2020) to learn and evaluate

ᴛ(ᴠ)c~PCFG.[2] To measure model performance, we resort to unlabeled corpus-level F1 (C-F1) and sentence-level F1 (S-F1), which are equivalent to recall in unsupervised grammar induction.

**Domain-specific vocabulary.** For each corpus, we keep the top 10,000 frequent words in the corresponding training set as the vocabulary. In training and test, tokens outside of the given vocabulary are treated as a special "<unk>" token (short for "unknown"). We share the vocabulary of each mixed domain among its subdomains, e.g., the 8 subdomains of Brown shares the vocabulary of the mixed domain Brown, similarly for Enweb.

**Test-time vocabulary.** At test time, we use domain-specific vocabulary rather than the training-time vocabulary (i.e., the MSCOCO vocabulary). The reasons for doing so include: (1) domain-specific vocabulary is likely to cover more target-domain words than the training-time vocabulary, and (2) this allows for fair comparison because the baseline c~PCFG also uses domain-specific vocabulary.

### 3.3 Main Results

**Lexical information is crucial for grammar induction despite the high unknown-word ratio.** Even with domain-specific vocabulary, we observe high proportions of unknown tokens in Brown and Enweb (i.e., above 30% of total words). Surprisingly, ᴛᴠc~PCFG still delivers decent transfer learning performance (i.e., above 40% S-F1). This leads us to hypothesize that the parser might not rely on lexical information at all. Instead, it may

---

[2] https://github.com/zhaoyanpeng/cpcfg.

| Model | NP | VP | PP | SBAR | ADJP | ADVP | C-F1 | S-F1 |
|---|---|---|---|---|---|---|---|---|
| Left Branching | 7.9 | 0.7 | 3.9 | 7.0 | 3.1 | 15.2 | 5.2 | 8.3 |
| Right Branching | 24.9 | **65.0** | 38.7 | **58.6** | 31.6 | 20.4 | 37.1 | 45.3 |
| Random Trees | $24.7_{\pm0.2}$ | $15.0_{\pm0.2}$ | $21.3_{\pm0.6}$ | $11.7_{\pm1.3}$ | $22.1_{\pm0.9}$ | $28.9_{\pm3.3}$ | $16.5_{\pm0.2}$ | $21.2_{\pm0.2}$ |
| c~PCFG [†] | **$75.0_{\pm3.1}$** | $31.9_{\pm16.2}$ | **$67.2_{\pm8.5}$** | $54.6_{\pm3.9}$ | **$39.7_{\pm7.8}$** | **$59.4_{\pm2.6}$** | **$47.8_{\pm4.4}$** | **$51.3_{\pm6.1}$** |
| ʟ1Oc~PCFG [†] | $63.3_{\pm1.8}$ | $25.5_{\pm23.5}$ | $53.7_{\pm6.7}$ | $36.2_{\pm7.9}$ | $28.2_{\pm8.9}$ | $40.2_{\pm3.1}$ | $38.3_{\pm6.2}$ | $42.8_{\pm8.9}$ |
| ᴛc~PCFG [*] | $34.7_{\pm8.3}$ | $28.9_{\pm5.9}$ | $38.8_{\pm10.5}$ | $34.2_{\pm3.8}$ | $26.3_{\pm2.3}$ | $33.3_{\pm2.6}$ | $27.5_{\pm1.7}$ | $33.7_{\pm1.8}$ |
| PERM | $22.9_{\pm0.2}$ | $22.6_{\pm3.4}$ | $24.1_{\pm1.9}$ | $17.3_{\pm2.2}$ | $21.5_{\pm3.0}$ | $24.7_{\pm0.8}$ | $18.7_{\pm1.4}$ | $25.4_{\pm1.8}$ |
| ᴛvc~PCFG [*] | $58.5_{\pm4.0}$ | $29.7_{\pm2.7}$ | $44.8_{\pm6.5}$ | $34.4_{\pm1.3}$ | $32.9_{\pm3.1}$ | $38.1_{\pm1.2}$ | $35.9_{\pm1.4}$ | $41.4_{\pm1.8}$ |
| PERM | $26.6_{\pm0.8}$ | $22.1_{\pm2.0}$ | $26.6_{\pm0.5}$ | $17.7_{\pm0.4}$ | $23.2_{\pm1.5}$ | $28.7_{\pm3.3}$ | $20.2_{\pm0.3}$ | $26.5_{\pm0.8}$ |
| Per-domain Performance of ᴛvc~PCFG | | | | | | | | |
| CF | $53.4_{\pm4.9}$ | $25.4_{\pm2.9}$ | $41.4_{\pm7.0}$ | $32.1_{\pm4.3}$ | $32.6_{\pm6.3}$ | $31.0_{\pm3.0}$ | $34.0_{\pm1.3}$ | $37.5_{\pm1.6}$ |
| CP | $64.1_{\pm3.0}$ | $31.9_{\pm3.8}$ | $49.0_{\pm8.2}$ | $41.1_{\pm1.9}$ | $34.5_{\pm3.2}$ | $39.7_{\pm7.3}$ | $37.7_{\pm1.6}$ | $44.1_{\pm2.1}$ |
| CN | $63.5_{\pm3.2}$ | $33.8_{\pm3.2}$ | $49.2_{\pm5.7}$ | $39.0_{\pm2.8}$ | $34.3_{\pm5.6}$ | $45.8_{\pm7.6}$ | $38.9_{\pm1.2}$ | $42.9_{\pm1.3}$ |
| CM | $61.4_{\pm5.5}$ | $36.4_{\pm2.9}$ | $49.6_{\pm5.5}$ | $39.5_{\pm4.5}$ | $44.8_{\pm8.6}$ | $50.0_{\pm10.2}$ | $40.0_{\pm1.0}$ | $46.3_{\pm1.3}$ |
| CG | $53.6_{\pm4.6}$ | $25.8_{\pm3.1}$ | $39.1_{\pm6.8}$ | $31.0_{\pm1.8}$ | $26.3_{\pm3.3}$ | $28.6_{\pm2.7}$ | $32.6_{\pm1.8}$ | $37.1_{\pm2.2}$ |
| CR | $52.0_{\pm3.3}$ | $24.0_{\pm2.9}$ | $39.0_{\pm6.7}$ | $25.9_{\pm3.2}$ | $26.8_{\pm3.0}$ | $34.5_{\pm7.4}$ | $31.8_{\pm0.8}$ | $35.0_{\pm1.4}$ |
| CK | $61.0_{\pm4.8}$ | $29.7_{\pm2.0}$ | $46.6_{\pm6.3}$ | $33.1_{\pm2.2}$ | $34.0_{\pm3.3}$ | $35.5_{\pm0.6}$ | $35.9_{\pm1.5}$ | $42.6_{\pm1.8}$ |
| CL | $62.2_{\pm3.9}$ | $32.6_{\pm1.6}$ | $46.4_{\pm6.3}$ | $39.7_{\pm3.4}$ | $35.3_{\pm1.7}$ | $40.5_{\pm4.2}$ | $36.7_{\pm1.4}$ | $41.3_{\pm2.0}$ |

Table 4: Parsing performance on Brown. [†] indicates the results obtained by running c~PCFG on Brown; [*] indicates the best models (w/ pre-trained GloVe word embeddings) that are trained on MSCOCO but evaluated on Brown. The 8 subdomains of Brown are (1) CF: popular lore, (2) CG: belles lettres, biography, memoires, etc., (3) CK: general fiction, (4) CL: mystery and detective fiction, (5) CM: science fiction, (6) CN: adventure and western fiction, (7) CP: romance and love story, and (8) CR: humor.

simply learn a heuristic composition strategy (e.g., a variation on the right branching baseline).

To test this hypothesis, we construct a permutation baseline, PERM. For sentences of the same length, we randomly exchange the trees of the sentences, inferred by the parser. After the exchange, a tree may not correspond to the associated sentence, i.e., the tree can be seen as being inferred without using the correct lexical information. If the parser does not exploit the lexical information, the performance of such baseline should be equal to that of the parser; accordingly, our hypothesis would be proven to be true.

In Table 3-5, we see PERM performs far worse than ᴛ(v)c~PCFG on all the test domains, disproving our hypothesis and demonstrating that ᴛvc~PCFG indeed relies on the token information.

vc~PCFG **benefits from pre-trained GloVe.** We run experiments on MSCOCO with pre-trained GloVe word embeddings (see Table 1). When trained on both images and text, ᴛvc~PCFG improves over vc~PCFG (+6.9% S-F1). But when trained only on text, ᴛc~PCFG lags far behind c~PCFG, i.e., using pre-trained GloVe leads to a reduction in performance.

We speculate that this is because domain-specific lexical information is important for grammar induction models. GloVe has been pre-trained on diverse text and may not best reflect lexical information relevant to the domain of MSCOCO captions (e.g., wrong senses and parts of speech), so ᴛc~PCFG underperforms c~PCFG.

But visual groundings are specific to a domain and could regularize a parser to capture domain-specific lexical information (Zhao and Titov, 2020), so ᴛvc~PCFG is less prone to the same issue as in ᴛc~PCFG; instead, it might be making the best of both visual groundings and pre-trained GloVe, so it outperforms vc~PCFG.

ᴛvc~PCFG **succeeds in proximate-domain transfer.** We train ᴛvc~PCFG on MSCOCO and evaluate it on Flickr without further training (see Table 2). Our transfer learning model achieves the best corpus- and sentence-level F1 scores on MSCOCO. When evaluated on Flickr, it outperforms c~PCFG (+20.8% S-F1; see Figure 1), so the improvements brought by visual groundings can transfer to similar text domains.

ᴛvc~PCFG **fails in remote-domain transfer.** We further evaluate pre-trained ᴛvc~PCFG on remote-domain text, including WSJ, Brown, and Enweb (see Table 3-5). On the whole, the trans-

| Model | NP | VP | PP | SBAR | ADJP | ADVP | C-F1 | S-F1 |
|---|---|---|---|---|---|---|---|---|
| Left Branching | 9.9 | 0.9 | 3.4 | 10.1 | 3.9 | 11.2 | 5.8 | 10.9 |
| Right Branching | 27.1 | **66.3** | 41.6 | **59.3** | 30.9 | 29.8 | 38.3 | **45.9** |
| Random Trees | $25.1_{\pm0.2}$ | $14.7_{\pm0.3}$ | $21.6_{\pm1.1}$ | $13.0_{\pm1.0}$ | $22.1_{\pm1.9}$ | $32.9_{\pm2.0}$ | $16.8_{\pm0.2}$ | $23.1_{\pm0.3}$ |
| C~PCFG [†] | $\mathbf{62.8}_{\pm2.6}$ | $25.5_{\pm10.4}$ | $\mathbf{53.5}_{\pm12.4}$ | $52.9_{\pm2.4}$ | $32.6_{\pm5.8}$ | $\mathbf{48.5}_{\pm9.1}$ | $\mathbf{39.7}_{\pm4.5}$ | $43.5_{\pm4.9}$ |
| L10C~PCFG [†] | $56.4_{\pm2.2}$ | $24.6_{\pm9.6}$ | $33.0_{\pm4.9}$ | $24.1_{\pm4.3}$ | $24.2_{\pm2.3}$ | $29.2_{\pm2.8}$ | $31.5_{\pm3.2}$ | $37.5_{\pm2.9}$ |
| TC~PCFG [*] | $34.9_{\pm6.8}$ | $28.0_{\pm7.0}$ | $41.1_{\pm10.2}$ | $34.2_{\pm4.2}$ | $27.4_{\pm1.7}$ | $38.3_{\pm5.2}$ | $27.6_{\pm2.0}$ | $34.3_{\pm2.2}$ |
| PERM | $23.9_{\pm0.9}$ | $22.3_{\pm4.6}$ | $25.7_{\pm1.5}$ | $19.2_{\pm3.3}$ | $24.0_{\pm2.1}$ | $29.1_{\pm2.9}$ | $19.4_{\pm1.8}$ | $26.9_{\pm2.3}$ |
| TVC~PCFG [*] | $55.0_{\pm3.9}$ | $28.6_{\pm3.6}$ | $45.4_{\pm6.1}$ | $34.9_{\pm0.4}$ | $\mathbf{35.1}_{\pm2.5}$ | $41.4_{\pm7.1}$ | $34.6_{\pm1.5}$ | $40.4_{\pm1.6}$ |
| PERM | $27.6_{\pm0.8}$ | $21.9_{\pm1.7}$ | $28.1_{\pm1.0}$ | $20.0_{\pm0.5}$ | $25.5_{\pm0.4}$ | $33.5_{\pm2.3}$ | $20.8_{\pm0.3}$ | $28.2_{\pm0.7}$ |
| Per-domain Performance of TVC~PCFG | | | | | | | | |
| Weblog | $51.2_{\pm4.8}$ | $26.8_{\pm3.4}$ | $40.4_{\pm5.4}$ | $31.0_{\pm3.7}$ | $30.8_{\pm5.3}$ | $42.0_{\pm4.5}$ | $32.7_{\pm0.8}$ | $38.0_{\pm0.6}$ |
| Answers | $58.9_{\pm3.4}$ | $29.3_{\pm2.9}$ | $50.7_{\pm5.2}$ | $35.1_{\pm1.9}$ | $37.7_{\pm6.3}$ | $43.4_{\pm6.9}$ | $34.8_{\pm1.4}$ | $39.0_{\pm2.0}$ |
| Email | $52.8_{\pm3.9}$ | $26.2_{\pm3.2}$ | $42.5_{\pm6.7}$ | $32.6_{\pm3.7}$ | $32.2_{\pm6.9}$ | $35.9_{\pm8.3}$ | $33.0_{\pm1.4}$ | $40.1_{\pm1.9}$ |
| Newsgroup | $50.9_{\pm2.7}$ | $27.3_{\pm3.5}$ | $41.2_{\pm7.8}$ | $33.7_{\pm1.4}$ | $29.2_{\pm5.6}$ | $36.9_{\pm4.6}$ | $33.7_{\pm1.8}$ | $36.9_{\pm2.4}$ |
| Reviews | $61.7_{\pm5.1}$ | $31.5_{\pm4.4}$ | $52.6_{\pm6.9}$ | $42.5_{\pm3.8}$ | $38.7_{\pm6.5}$ | $42.6_{\pm5.5}$ | $37.9_{\pm1.3}$ | $44.9_{\pm1.6}$ |

Table 5: Parsing performance on Enweb. [†] indicates the results obtained by running C~PCFG on Enweb; [*] indicates the best models (w/ pre-trained GloVe word embeddings) that are trained on MSCOCO but evaluated on Enweb.

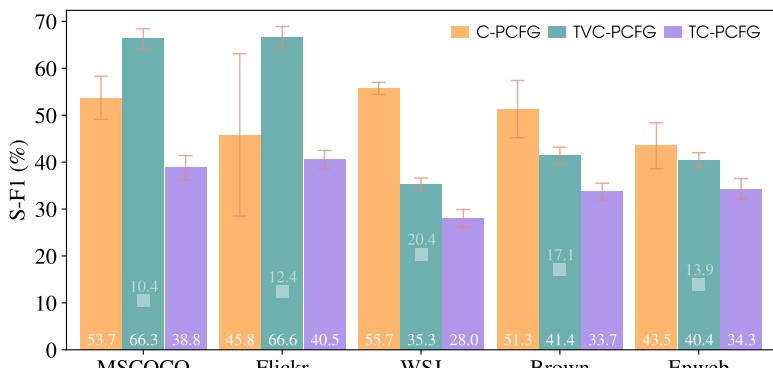

Figure 1: S-F1 numbers on different target domains. C~PCFG is trained only on the text data of each domain's training set. TVC~PCFG is our transfer learning model and TC~PCFG is the transfer learning model that is trained without using visual groundings. The squares indicate the average length of the test sentences of each domain.

fer learning model TVC~PCFG underperforms C~PCFG, which is trained individually on the training set of each target domain (see Figure 1). We observe the largest S-F1 gap between TVC~PCFG and C~PCFG on WSJ (-20.4%) and the smallest S-F1 gap on Enweb (-3.1%). This may be because of differences in language register. Both WSJ and Enweb are different from MSCOCO at a lexical level, but Enweb, consisting of web text, contains informal language which is likely to be structurally similar to that of captions.

Regarding model performance on subdomains, we observe similar trends as we see on mixed domains. Specifically, on the subdomains of both Brown and Enweb, C~PCFG performs best, and TVC~PCFG outperforms TC~PCFG (see Figure 10a in Appendix).

**Remote-domain training is helpful.** Since the average lengths of WSJ, Brown, and Enweb training sentences are higher than that of MSCOCO

training captions, to allow for fair comparison, for each target domain, we further train C~PCFG individually on the training sentences of the length below 10.5, the average length of MSCOCO training captions. We dub this model L10C~PCFG.[3]

Surprisingly, on WSJ and Brown, though the sentences used for training L10C~PCFG are shorter than 10.5 tokens, L10C~PCFG surpasses TVC~PCFG by 12.9% and 1.4% S-F1, respectively (see Figure 2). On Enweb, while TVC~PCFG beats L10C~PCFG (+2.9% S-F1), it does not always outperform L10C~PCFG on every run, despite that the average length of the Enweb sentences used for training L10C~PCFG is only 5 (*cf.* 10.5 tokens).

Across the remote-domain test sets, we also observe that the longer the sentences are used for training C~PCFG, the better the performance is. For example, C~PCFG always surpasses L10C~PCFG. Interestingly, without considering the "domain"

---

[3]Alternatively, we can choose a length cutoff that results in a subset that has a similar average length to MSCOCO.

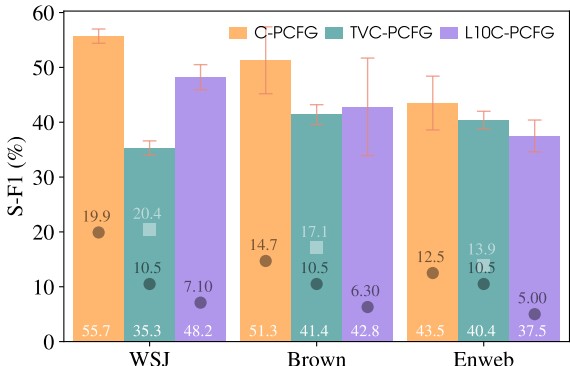

Figure 2: C~PCFG is trained on sentences shorter than 41 tokens, L10C~PCFG is trained on sentences shorter than 11 tokens, and TVC~PCFG is our transfer learning model. The circles represent the average length of the sentences for training each model; the squares indicate the average length of the test sentences of each domain.

variable, the improvement of L10C~PCFG over TVC~PCFG becomes larger as the average length of the sentences used for training L10C~PCFG increases: -2.9% < +1.4% < +12.9% S-F1 with 5.0 < 6.3 < 7.1 tokens for Enweb, Brown, and WSJ, respectively.

With regard to model performance on subdomains, again, we observe similar trends as we see on mixed domains. Specifically, on the subdomains of both Brown and Enweb, C~PCFG performs best, and TVC~PCFG underperforms L10C~PCFG on the subdomains of Brown but outperforms L10C~PCFG on the subdomains of Enweb (see Figure 5).

### 3.4 Data Analysis

To investigate the relationship between transferability and source-target domain similarity, we look into three dimensions of variation (we call them "factors") that we hypothesize to influence transfer learning performance: variation in terms of distributions of phrasal labels, grammar rules, and words. For each of these three factors, we compute overlap rates between the MSCOCO training set and each target test set. We present two types of overlap rates: type-level and instance-level. The type-level overlap rate is the proportion of all the types of a factor in a test set that is covered by the MSCOCO training set, and the instance-level overlap rate is the proportion of all the instances of a factor in a test set that is covered by the MSCOCO training set (see Figure 3).

**Flickr is most similar to MSCOCO.** In Figure 3, we observe a large overlap between the MSCOCO

training set and the Flickr test set for all three factors in terms of both type-level and instance-level overlap rates (i.e., above 88% on the type level and above 99% on the instance level), so unsurprisingly, TVC~PCFG transfers to Flickr.

**WSJ is least similar to MSCOCO.** For the WSJ test set, the type-level overlap rates for grammar rules and tokens are below 50%, though the overlapped factor types cover the majority of the test data (i.e., 86.7% of rule occurrences and 79.7% of token occurrences). This suggests that WSJ is more diverse than MSCOCO in terms of syntactic structures and lexicons, and accounts for the mediocre transfer learning performance of TVC~PCFG on WSJ. While Brown and Enweb have slightly larger overlap rates than WSJ (i.e., around 55%), their overlaps with MSCOCO are far smaller than Flickr (55% versus 88%).

In terms of phrasal-label overlap, over 87% of each remote-domain test set is covered by the MSCOCO training set. This is unsurprising because the parser used for parsing MSCOCO captions is pre-trained on the WSJ of the Penn Treebank, and both Brown and Enweb follow the annotation specifications of the Penn Treebank.

**The larger instance-level overlaps, the better.** Our results show that L10C~PCFG performs surprisingly well (48.2% S-F1) on the WSJ test set while TVC~PCFG does not (35.3% S-F1). To investigate the reasons, we compare their training sets: MSCOCO and WSJ-L10 (i.e., WSJ training sentences shorter than 11 tokens) (see Figure 4). Since WSJ-L10 consists of only around 6,000 sentences (*cf.* 413,915 MSCOCO captions), unsurprisingly, the type-level overlap rates for grammar rules and words between WSJ-L10 and the WSJ test set are slightly lower than those between MSCOCO and the WSJ test set (e.g., -0.5% for words), but the overlaps cover higher proportions of rule and word occurrences in the WSJ test set (e.g., +6.3% for words), possibly explaining the better performance of L10C~PCFG (results on Brown and Enweb can be found in Figure 11 in Appendix).

**The lexicon coverage is the most important factor.** In fact, the higher type-level overlap rate for grammar rules (+4%) between MSCOCO and the WSJ test set is largely contributed by the non-lexical rule overlap (see overlap rates broken down by rule types in Figure 4). In terms of the lexical rule overlap, WSJ-L10 has a slightly smaller

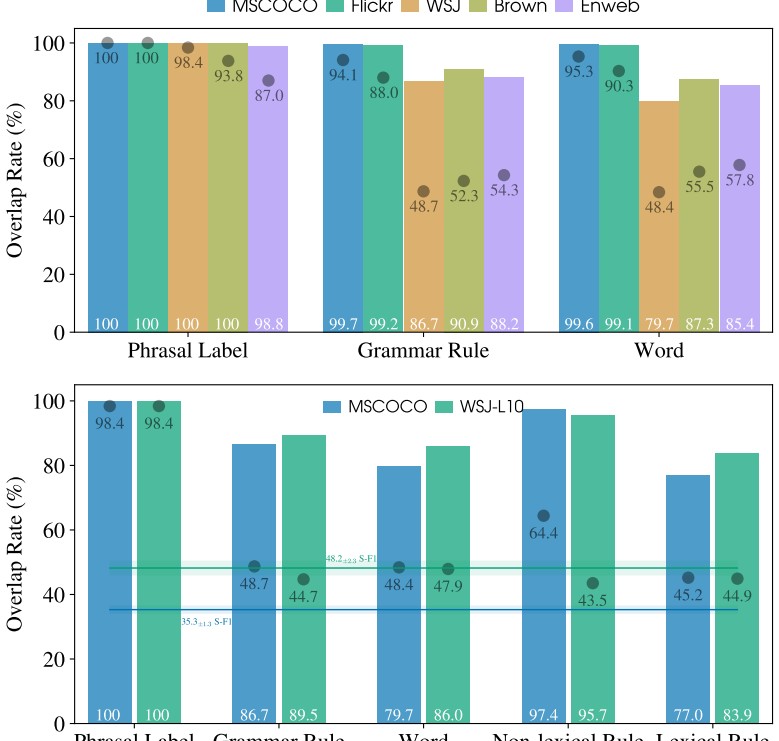

Figure 3: Overlap rates between the MSCOCO training set and five target test sets: MSCOCO, Flickr, WSJ, Brown, and Enweb. The bars (or the numbers in white) indicate the instance-level overlap rates of phrasal labels, grammar rules, and words, while the circles indicate type-level overlap rates.

Figure 4: Overlap rates of the WSJ test set with the MSCOCO training set and with the WSJ-L10 training set. The bars (or the numbers in white) indicate the instance-level overlap rates, while the circles indicate type-level overlap rates. The horizontal lines indicate test performance, i.e., S-F1 (%).

overlap rate (-0.3%) but covers a much higher portion of lexical rule occurrences (+6.9%). Given the better performance of L10C~PCFG, we conclude that lexicon overlap is the most important factor in the transferability of VC~PCFG on WSJ. Note that a similar finding from Zhao and Titov (2021) suggests that lexical rules, out of all the rule types, have the greatest influence on the performance of un-grounded C~PCFG. We conjecture that the finding also applies to VC~PCFG.

To quantify the correlation between lexicon overlap and the performance of TVC~PCFG, we further compute Spearman's rank correlation coefficient on mixed domains and their subdomains (see Figure 5). We find that (1) S-F1 and lexical rule overlap are positively correlated with a coefficient of 0.59 ($p$-value = 0.02); and (2) S-F1 and unknown token rate are negatively correlated with a coefficient of 0.69 ($p$-value = 0.004), once again confirming our conclusion.

### 3.5 Error Analysis

Our analysis so far has suggested that overlap in terms of words and lexical rules are critical factors in the transferability of TVC~PCFG. But due to the domain difference, some of the target-domain words are inevitably classified as unknown words due to their low frequencies. To investigate how this influences parsing performance, we further

conduct error analysis. Specifically, given the inferred parse of a sentence, we count correctly recognized gold phrases and unrecognized gold phrases. To correlate with unknown words, we instead perform the counting for a set of sentences that have the same length and contain the same number of unknown tokens, then we compute the ratio of the number of correctly recognized gold phrases to the number of unrecognized gold phrases. But, the number of sentences is not generally evenly distributed across the sentence length. To obtain sufficiently reliable statistics, we divide sentences into buckets, where each bucket contains sentences that fall into a consecutive length range.

**Error rate versus the number of unseen tokens.** Overall, the ratio decreases as the sentence length increases, i.e., the chance of making mistakes increases (see Figure 6). When the sentence length is above 17, the chance of failure becomes higher than that of success. But, as the number of unknown tokens increases, we do not observe similar ratio changes across sentence length buckets. This might be because of insufficient statistics for a specific number of unknown tokens.

To measure the correlation between the ratio and the sentence length, we compute Spearman's Rank correlation coefficient for a particular unknown token number. We chose unknown-token numbers 0, 1, and 2 because they are covered by all sentence

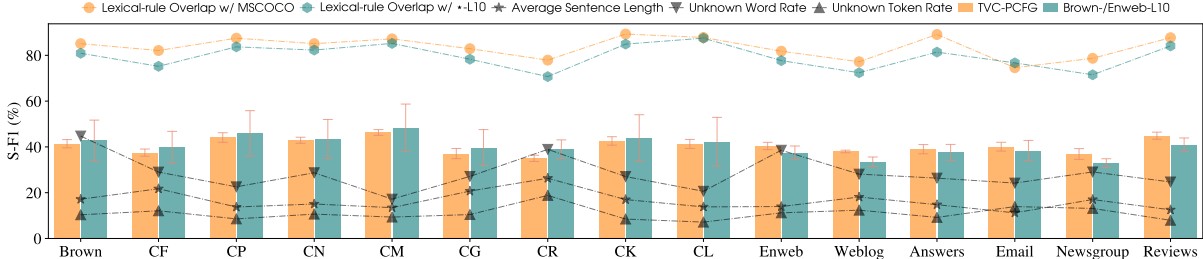

Figure 5: Transfer learning performance on the subdomains of Brown and Enweb. The lexical-rule overlap rates are computed on the instance level. The average sentence lengths are computed on the test set of each domain.

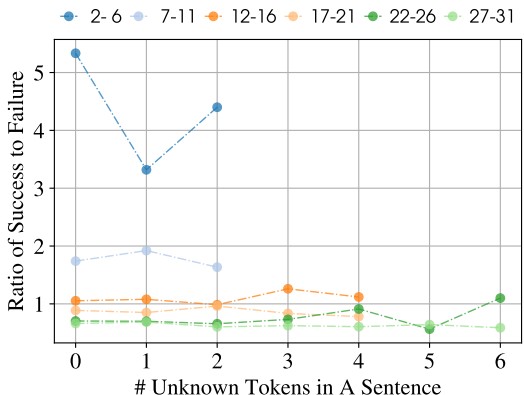

Figure 6: The ratio of success to failure for each sentence length bucket.

| # "<unk>" | 0 | 1 | 2 |
|-----------|------|------|------|
| $\rho$ | -0.9170 | -0.833 | -0.983 |
| $p$-value | 0.0005 | 0.005 | 0 |

Table 6: Spearman's Rank correlation between the ratio and the sentence length for an unknown-token number.

buckets (see Figure 6). The correlation test results confirm that our observation about the negative correlation between the ratio and the sentence length is statistically reliable (see Table 6).

## 4 Conclusion and Future Work

We have presented a simple approach that enables VC~PCFG to transfer to text domains beyond the training domain. Our approach relies on pre-trained word embeddings and does not require training on the target domain. We empirically find that our TVC~PCFG is able to transfer to similar text domains but struggles to transfer to remote-domain text. Through data and result analysis, we find that the lexicon overlap between the source domain and the target domain has a great effect on the transferability of TVC~PCFG.

In the future, we would like to explore alternative transfer learning settings. For example, when training sentences from the target domain are available, we can additionally optimize the LM objective on them, apart from optimizing the full objective function on source-domain image-text pairs. Moreover, we may consider a few-shot transfer learning setting and learn VC~PCFG only on a few target-domain sentences.

## Limitations

We acknowledge these limitations of our work:

**English-specific transfer learning.** We have so far focused on transferring VC~PCFG across genres of text in English, primarily because, for English, there are many human-annotated treebank resources allowing for reliable evaluation, but this also implies that our findings may be limited to English, though our approach is applicable to other languages. Specifically, we can replace pre-trained GloVe embeddings with pre-trained multilingual word embeddings (Smith et al., 2017) and achieve cross-lingual transfer.

**Model variance.** Both Kim et al. (2019) and Zhao and Titov (2021) have noticed the variance issue of C~PCFG. In our work, we also observed high variances, for example, on Flickr and Brown. We conjecture that this is an issue of all PCFG-based models (Petrov, 2010), not just the neural extensions used in this work. To have a better understanding of this issue, we think that more thorough studies on the stability of PCFG-based grammar-induction models are needed.

## Acknowledgements

Ivan Titov is partially supported by the Dutch National Science Foundation (NWO Vici VI.C.212.053).

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

## Abstract

In this Appendix, we elaborate on (1) target-domain word embedding selection (Section A); (2) statistics of the Brown and Enweb corpora (Section B); (3) more experimental results (Section C).

## A  Word Embedding Selection

To evaluate our transfer learning model, we could directly apply the learned model to the target domain, which is equivalent to using the training-time vocabulary (i.e., the MSCOCO vocabulary), but due to domain differences, the training-time vocabulary may cover fewer target-domain words than the target-domain vocabulary (i.e., domain-specific vocabulary) and lead to deteriorated performance. Thus, we further study different ways of selecting the embeddings of the words in the target domain.

• *Direct.* We directly use the training-time vocabulary, i.e., for each word in the test set, it is treated as an unknown word if it does not appear in the training-time vocabulary (see Figure 7 for an illustration).

• *Random.* We use the vocabulary of the target domain. Each word in the test set is treated as an unknown word if it is not covered by the target-domain vocabulary; otherwise, if it does not have an embedding in GloVe, we use random initialization (see Figure 7 for an illustration).

• *Unknown.* We use the vocabulary of the target domain. Each word in the test set is treated as an unknown word if it is not covered by the target-domain vocabulary or it does not have an embedding in GloVe. *Unknown* differs from *Random* in that it treats the small set of words that uses random embeddings in *Random* as a "<unk>" token.

• *Standard.* We use the vocabulary of the target domain. For each word in the test set, it is treated as an unknown word if it is not covered by the target-domain vocabulary; otherwise, if is not covered by the training-time vocabulary and the GloVe vocabulary, we use random initialization.[4] Compared with *Random*, for the small set of words that use random embeddings in *Random*, *Standard* further seeks their embeddings from the learned TVC~PCFG.

---

[4] A target-domain word may be covered by the training-time vocabulary but is not covered by the GloVe vocabulary. If so, we use the learned embedding from the transfer learning model, i.e., in *Standard*, we always try to find a pre-trained word embedding first.

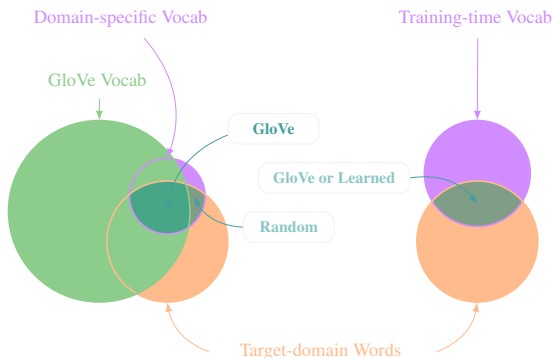

Figure 7: Embedding selection with the *Random* strategy (left) and the *Direct* strategy (right).

Surprisingly, we do not observe significant S-F1 differences between the four embedding selection methods (see Figure 8). To gain a thorough understanding of this phenomenon, we specifically investigate and compare the *Random* and *Direct* strategies. For the *Random* strategy, we find that there are only tens of words are assigned random embeddings, i.e., almost all of the words that are covered by the target-domain vocabulary have a GloVe embedding. Thus, the major difference between the two strategies lies in unknown words. Since the "<unk>" token appears in GloVe, the two strategies share the embedding of "<unk>", so we can further narrow down the difference to the number of unknown words.

To spell out the difference, for each test domain, we compute the proportions of unknown words on the word (i.e., word type) level and on the token level, which are respectively denoted by circles and squares in Figure 9. We can see that *Random* results in larger proportions of unseen words than *Direct* on all the domains except Flickr, but the token-level proportion differences are very small, e.g., they are less than 0.5% on MSCOCO, Flickr, and Brown. This may explain why the two selection strategies lead to similar parsing performance on all the test domains.

Though *Direct* tends to perform slightly better than *Random* (+0.1% mean S-F1) across the test domains, we use *Random* as the default strategy in this work because it shares the vocabulary with the baseline parser (i.e., C~PCFG) trained solely on the target-domain text and allows for fair comparison.

## B  Data Statistics

We present statistics of Brown and Enweb in Table 7 and 8, respectively.

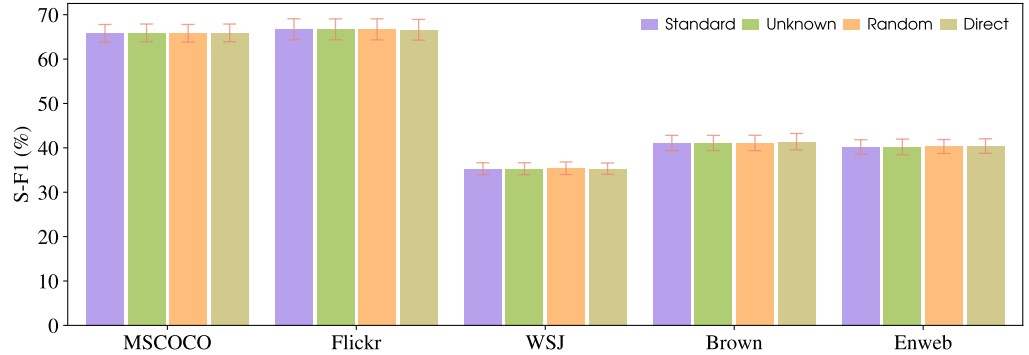

Figure 8: S-F1 numbers with different ways of selecting the embeddings of target-domain words.

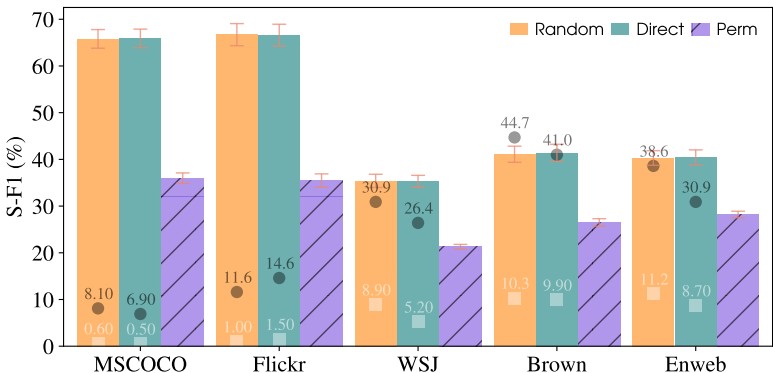

Figure 9: S-F1 numbers with different ways of selecting the embeddings of target-domain words. The circles and squares indicate the proportions of words and tokens in the test set that are treated as "<unk>", respectively (see Appendix A for the *Random* and *Direct* strategies and Section 3.2 for the PERM baseline).

## C   Experimental Results

• We present the performance of ʟ10ᴄ~PCFG and ᴛᴄ~PCFG on subdomains of Brown and Enweb in Table 9 and 10, respectively.

• Figure 10 visualizes the performance of ᴄ~PCFG, ʟ10ᴄ~PCFG, ᴛᴠᴄ~PCFG, and ᴛᴄ~PCFG on all the test domains.

• Figure 11 plots overlap rates of each remote-domain test set with the corresponding remote-domain training sentences (shorter than 11 tokens) and with the MSCOCO training set.

• Figure 12 visualizes the ratios of success to failure on WSJ, Brown, and Enweb.

| Split | CF | CG | CK | CL | CM | CN | CP | CR | All (Brown) |
|---|---|---|---|---|---|---|---|---|---|
| train | 2191 | 2324 | 2708 | 2745 | 615 | 3267 | 2801 | 648 | 17299 |
| dev | 507 | 461 | 570 | 518 | 115 | 599 | 543 | 164 | 3477 |
| test | 466 | 494 | 603 | 451 | 151 | 549 | 598 | 155 | 3467 |
| | | | | File ID Splits | | | | | |
| train | 1-22 | 1-25 | 1-19 | 1-18 | 1-4 | 1-21 | 1-20 | 1-6 | |
| dev | 23-27 | 26-31 | 20-23 | 19-21 | 5-5 | 22-25 | 21-25 | 7-7 | |
| test | 28-32 | 32-36 | 24-29 | 22-24 | 6-6 | 26-29 | 26-29 | 8-9 | |

Table 7: Nine subdomains of the Brown corpus of Penn Treebank (Marcus et al., 1999). CF: popular lore. CG: belles lettres, biography, memoires, etc. CK: general fiction. CL: mystery and detective fiction. CM: science fiction. CN: adventure and western fiction. CP: romance and love story. CR: humor.

| Split | Answers | Email | Newsgroup | Reviews | Weblog | All (Enweb) |
|---|---|---|---|---|---|---|
| train | 2353 | 3362 | 1648 | 2565 | 1451 | 11379 |
| dev | 565 | 767 | 368 | 622 | 245 | 2567 |
| test | 569 | 759 | 371 | 626 | 334 | 2659 |

Table 8: Five subdomains of the English Web Treebank (Bies et al., 2012)

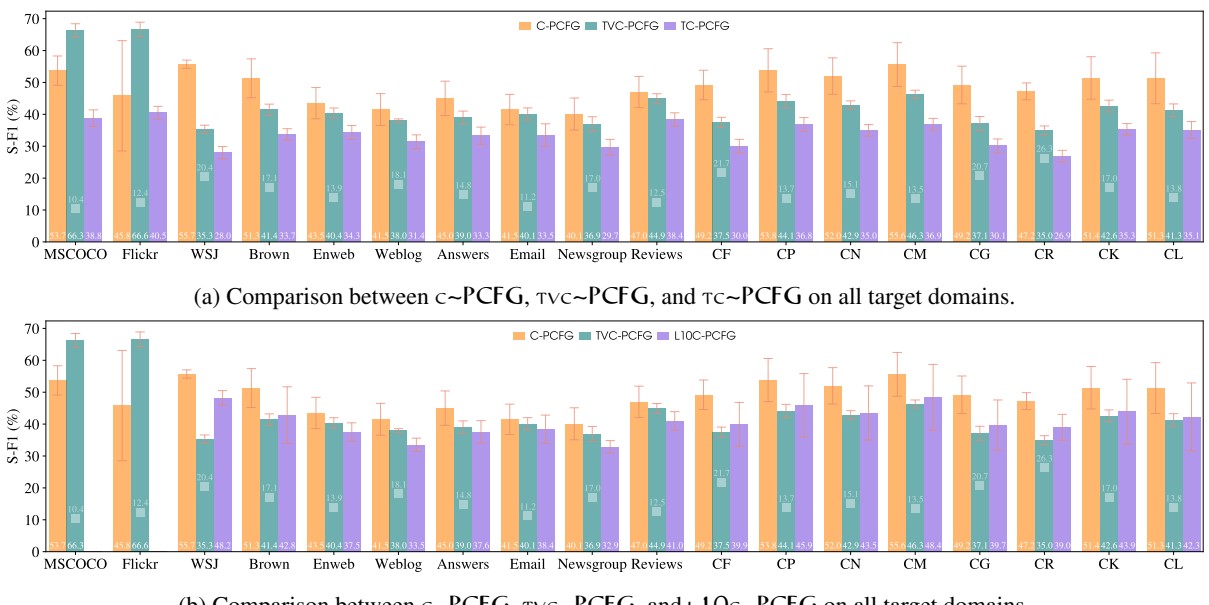

(a) Comparison between C~PCFG, TVC~PCFG, and TC~PCFG on all target domains.

(b) Comparison between C~PCFG, TVC~PCFG, and L10C~PCFG on all target domains.

Figure 10: C~PCFG is trained on sentences shorter than 41 tokens, L10C~PCFG is trained on sentences shorter than 11 tokens, TVC~PCFG is our transfer learning model, and TC~PCFG is the transfer learning model that is trained without using visual groundings. The squares indicate the average length of the test sentences of each domain.

| Model | NP | VP | PP | SBAR | ADJP | ADVP | C-F1 | S-F1 |
|---|---|---|---|---|---|---|---|---|
| c~PCFG [†] | $75.0_{\pm3.1}$ | $31.9_{\pm16.2}$ | $67.2_{\pm8.5}$ | $54.6_{\pm3.9}$ | $39.7_{\pm7.8}$ | $59.4_{\pm2.6}$ | $47.8_{\pm4.4}$ | $51.3_{\pm6.1}$ |
| L10c~PCFG [†] | $63.3_{\pm1.8}$ | $25.5_{\pm23.5}$ | $53.7_{\pm6.7}$ | $36.2_{\pm7.9}$ | $28.2_{\pm8.9}$ | $40.2_{\pm3.1}$ | $38.3_{\pm6.2}$ | $42.8_{\pm8.9}$ |
| Per-domain Performance of L10c~PCFG | | | | | | | | |
| CF | $58.4_{\pm2.6}$ | $24.0_{\pm23.5}$ | $49.5_{\pm6.7}$ | $31.5_{\pm9.2}$ | $31.2_{\pm9.0}$ | $31.7_{\pm5.5}$ | $37.3_{\pm5.5}$ | $39.9_{\pm6.9}$ |
| CP | $67.4_{\pm1.5}$ | $28.2_{\pm26.6}$ | $58.6_{\pm6.6}$ | $43.9_{\pm8.0}$ | $33.9_{\pm14.5}$ | $43.8_{\pm3.7}$ | $40.2_{\pm7.5}$ | $45.9_{\pm9.9}$ |
| CN | $68.0_{\pm2.4}$ | $26.3_{\pm24.0}$ | $58.3_{\pm5.1}$ | $35.6_{\pm7.2}$ | $30.2_{\pm10.5}$ | $45.3_{\pm7.1}$ | $39.7_{\pm6.5}$ | $43.5_{\pm8.5}$ |
| CM | $71.3_{\pm1.4}$ | $30.1_{\pm26.9}$ | $66.0_{\pm13.1}$ | $43.2_{\pm15.1}$ | $32.3_{\pm19.1}$ | $28.1_{\pm6.2}$ | $44.5_{\pm8.1}$ | $48.4_{\pm10.3}$ |
| CG | $59.1_{\pm1.7}$ | $23.3_{\pm21.0}$ | $47.6_{\pm6.3}$ | $32.3_{\pm8.1}$ | $24.8_{\pm8.6}$ | $36.6_{\pm4.9}$ | $35.9_{\pm5.5}$ | $39.7_{\pm7.9}$ |
| CR | $58.3_{\pm1.9}$ | $22.6_{\pm18.2}$ | $48.3_{\pm8.5}$ | $29.2_{\pm5.1}$ | $22.3_{\pm3.1}$ | $42.2_{\pm5.9}$ | $35.8_{\pm3.3}$ | $39.0_{\pm4.1}$ |
| CK | $65.7_{\pm1.5}$ | $25.1_{\pm22.6}$ | $55.4_{\pm6.6}$ | $36.2_{\pm6.9}$ | $25.2_{\pm8.2}$ | $44.4_{\pm4.0}$ | $38.3_{\pm6.3}$ | $43.9_{\pm10.2}$ |
| CL | $68.3_{\pm1.5}$ | $26.9_{\pm25.8}$ | $59.4_{\pm6.8}$ | $44.1_{\pm8.9}$ | $29.8_{\pm9.2}$ | $40.1_{\pm3.5}$ | $39.3_{\pm8.3}$ | $42.3_{\pm10.6}$ |
| TC~PCFG [*] | $34.7_{\pm8.3}$ | $28.9_{\pm5.9}$ | $38.8_{\pm10.5}$ | $34.2_{\pm3.8}$ | $26.3_{\pm2.3}$ | $33.3_{\pm2.6}$ | $27.5_{\pm1.7}$ | $33.7_{\pm1.8}$ |
| PERM | $22.9_{\pm0.2}$ | $22.6_{\pm3.4}$ | $24.1_{\pm1.9}$ | $17.3_{\pm2.2}$ | $21.5_{\pm3.0}$ | $24.7_{\pm0.8}$ | $18.7_{\pm1.4}$ | $25.4_{\pm1.8}$ |
| Per-domain Performance of TC~PCFG | | | | | | | | |
| CF | $32.6_{\pm7.2}$ | $26.8_{\pm7.0}$ | $38.3_{\pm9.9}$ | $33.3_{\pm2.6}$ | $21.0_{\pm4.4}$ | $29.2_{\pm5.2}$ | $26.6_{\pm1.9}$ | $30.0_{\pm2.2}$ |
| CP | $38.5_{\pm8.5}$ | $31.9_{\pm6.7}$ | $41.1_{\pm12.2}$ | $40.7_{\pm3.8}$ | $28.6_{\pm4.3}$ | $42.1_{\pm7.4}$ | $29.7_{\pm2.0}$ | $36.8_{\pm2.2}$ |
| CN | $38.3_{\pm8.8}$ | $32.4_{\pm6.3}$ | $40.7_{\pm10.7}$ | $36.3_{\pm5.0}$ | $26.9_{\pm1.9}$ | $44.3_{\pm9.8}$ | $29.5_{\pm1.4}$ | $35.0_{\pm1.8}$ |
| CM | $36.9_{\pm10.8}$ | $34.2_{\pm8.7}$ | $44.6_{\pm13.9}$ | $39.5_{\pm6.2}$ | $32.3_{\pm11.5}$ | $40.6_{\pm6.2}$ | $31.9_{\pm1.7}$ | $36.9_{\pm1.8}$ |
| CG | $32.3_{\pm6.6}$ | $25.1_{\pm6.6}$ | $35.1_{\pm9.7}$ | $31.0_{\pm4.2}$ | $19.1_{\pm2.7}$ | $27.2_{\pm3.2}$ | $25.0_{\pm1.9}$ | $30.1_{\pm2.2}$ |
| CR | $29.3_{\pm6.7}$ | $23.5_{\pm5.0}$ | $35.6_{\pm13.3}$ | $29.4_{\pm6.4}$ | $26.8_{\pm3.5}$ | $23.3_{\pm4.3}$ | $23.6_{\pm2.1}$ | $26.9_{\pm1.8}$ |
| CK | $35.5_{\pm9.5}$ | $28.5_{\pm5.2}$ | $39.6_{\pm11.9}$ | $32.2_{\pm4.0}$ | $30.7_{\pm2.5}$ | $28.7_{\pm7.0}$ | $27.5_{\pm1.6}$ | $35.3_{\pm1.8}$ |
| CL | $39.4_{\pm8.8}$ | $31.7_{\pm5.9}$ | $42.1_{\pm9.2}$ | $38.4_{\pm7.7}$ | $26.0_{\pm7.3}$ | $33.7_{\pm7.0}$ | $29.9_{\pm2.5}$ | $35.1_{\pm2.6}$ |

Table 9: Parsing performance on Brown. [†] indicates the results obtained by running c~PCFG on Brown; [*] indicates the best models (w/ pre-trained GloVe word embeddings) trained on MSCOCO but evaluated on Brown.

| Model | NP | VP | PP | SBAR | ADJP | ADVP | C-F1 | S-F1 |
|---|---|---|---|---|---|---|---|---|
| c~PCFG [†] | $62.8_{\pm2.6}$ | $25.5_{\pm10.4}$ | $53.5_{\pm12.4}$ | $52.9_{\pm2.4}$ | $32.6_{\pm5.8}$ | $48.5_{\pm9.1}$ | $39.7_{\pm4.5}$ | $43.5_{\pm4.9}$ |
| L10c~PCFG [†] | $56.4_{\pm2.2}$ | $24.6_{\pm9.6}$ | $33.0_{\pm4.9}$ | $24.1_{\pm4.3}$ | $24.2_{\pm2.3}$ | $29.2_{\pm2.8}$ | $31.5_{\pm3.2}$ | $37.5_{\pm2.9}$ |
| Per-domain Performance of L10c~PCFG | | | | | | | | |
| Weblog | $52.6_{\pm1.4}$ | $20.0_{\pm10.1}$ | $29.6_{\pm4.8}$ | $18.6_{\pm7.1}$ | $13.5_{\pm5.9}$ | $31.2_{\pm5.4}$ | $29.4_{\pm2.7}$ | $33.5_{\pm2.1}$ |
| Answers | $60.0_{\pm1.9}$ | $27.2_{\pm9.3}$ | $39.1_{\pm5.4}$ | $24.2_{\pm4.1}$ | $29.1_{\pm3.3}$ | $26.2_{\pm4.1}$ | $32.1_{\pm4.3}$ | $37.6_{\pm3.5}$ |
| Email | $54.8_{\pm3.7}$ | $24.4_{\pm9.5}$ | $32.0_{\pm5.1}$ | $26.2_{\pm6.3}$ | $20.1_{\pm5.3}$ | $17.2_{\pm6.0}$ | $31.0_{\pm2.9}$ | $38.4_{\pm4.4}$ |
| Newsgroup | $53.2_{\pm1.9}$ | $20.7_{\pm9.7}$ | $29.6_{\pm4.4}$ | $20.3_{\pm5.9}$ | $15.7_{\pm4.2}$ | $38.1_{\pm6.7}$ | $30.4_{\pm2.4}$ | $32.9_{\pm1.9}$ |
| Reviews | $62.1_{\pm2.4}$ | $27.5_{\pm10.5}$ | $35.8_{\pm6.2}$ | $28.5_{\pm4.8}$ | $30.1_{\pm1.8}$ | $37.8_{\pm4.4}$ | $34.0_{\pm3.8}$ | $41.0_{\pm2.9}$ |
| TC~PCFG [*] | $34.9_{\pm6.8}$ | $28.0_{\pm7.0}$ | $41.1_{\pm10.2}$ | $34.2_{\pm4.2}$ | $27.4_{\pm1.7}$ | $38.3_{\pm5.2}$ | $27.6_{\pm2.0}$ | $34.3_{\pm2.2}$ |
| PERM | $23.9_{\pm0.9}$ | $22.3_{\pm4.6}$ | $25.7_{\pm1.5}$ | $19.2_{\pm3.3}$ | $24.0_{\pm2.1}$ | $29.1_{\pm2.9}$ | $19.4_{\pm1.8}$ | $26.9_{\pm2.3}$ |
| Per-domain Performance of TC~PCFG | | | | | | | | |
| Weblog | $30.6_{\pm6.2}$ | $27.4_{\pm7.3}$ | $37.3_{\pm9.5}$ | $34.7_{\pm6.1}$ | $21.2_{\pm5.9}$ | $43.8_{\pm12.8}$ | $26.0_{\pm2.0}$ | $31.4_{\pm2.2}$ |
| Answers | $38.5_{\pm5.7}$ | $28.3_{\pm6.5}$ | $44.5_{\pm11.5}$ | $34.2_{\pm3.2}$ | $30.3_{\pm3.2}$ | $39.7_{\pm8.4}$ | $28.5_{\pm2.4}$ | $33.3_{\pm2.7}$ |
| Email | $34.2_{\pm5.6}$ | $24.5_{\pm7.8}$ | $39.0_{\pm8.5}$ | $30.7_{\pm6.7}$ | $26.5_{\pm6.4}$ | $28.9_{\pm8.2}$ | $26.0_{\pm2.1}$ | $33.5_{\pm3.5}$ |
| Newsgroup | $31.3_{\pm6.7}$ | $25.8_{\pm7.0}$ | $39.8_{\pm10.5}$ | $34.0_{\pm8.1}$ | $21.6_{\pm2.8}$ | $32.1_{\pm13.7}$ | $26.1_{\pm2.0}$ | $29.7_{\pm2.5}$ |
| Reviews | $40.7_{\pm8.0}$ | $31.8_{\pm7.7}$ | $46.1_{\pm11.7}$ | $39.6_{\pm4.7}$ | $31.4_{\pm1.9}$ | $43.6_{\pm8.2}$ | $31.0_{\pm2.4}$ | $38.4_{\pm2.1}$ |

Table 10: Parsing performance on Enweb. [†] indicates the results obtained by running c~PCFG on Enweb; [*] indicates the best models (w/ pre-trained GloVe word embeddings) trained on MSCOCO but evaluated on Enweb.

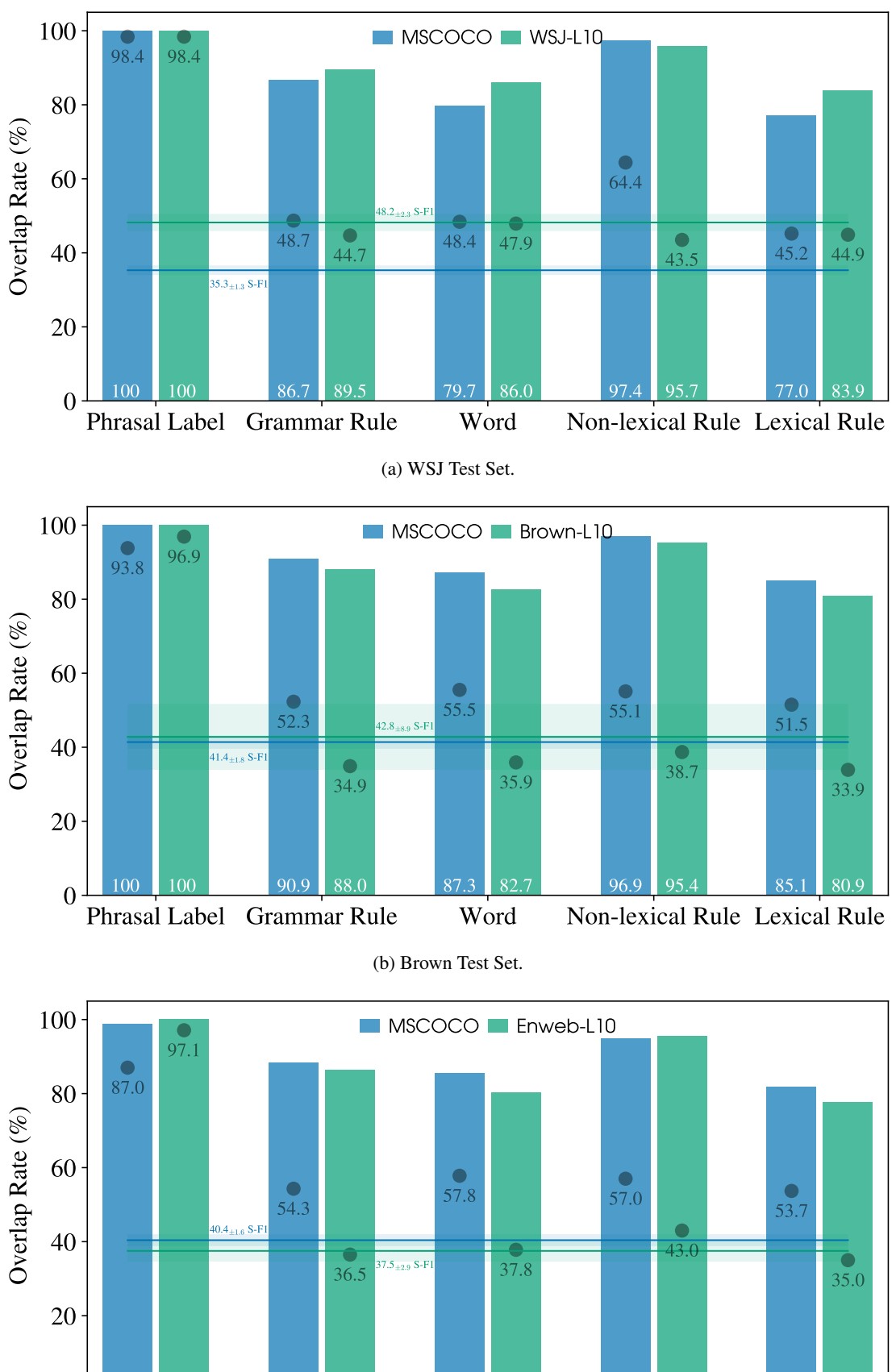

Figure 11: Overlap rates of the WSJ/Brown/Enweb test set with the MSCOCO training set and with the WSJ-L10/Brown-L10/Enweb-L10 training set.

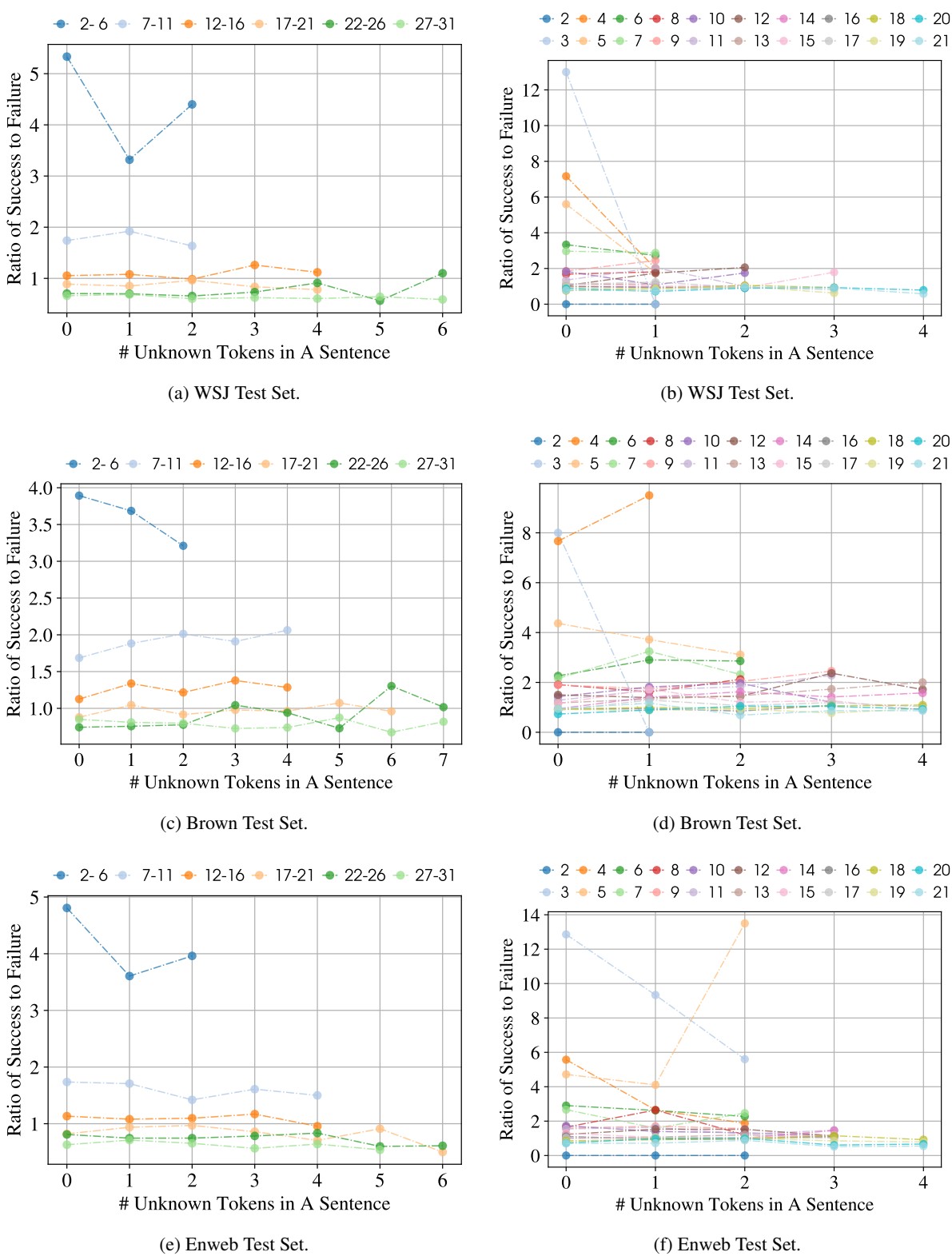

Figure 12: The ratio of success to failure for each sentence length bucket (left) and for individual sentence lengths (right). The $Y$ axis at zero indicates that TVC~PCFG makes zero mistakes.