# OpenReview forum: "On the Transferability of Visually Grounded PCFGs"
_EMNLP/2023/Conference — EMNLP 2023 Findings_

### Official Review · Reviewer_qPWB · 2023-08-05

**Soundness:** 3

**Excitement:**

3: Ambivalent: It has merits (e.g., it reports state-of-the-art results, the idea is nice), but there are key weaknesses (e.g., it describes incremental work), and it can significantly benefit from another round of revision. However, I won't object to accepting it if my co-reviewers champion it.

**Paper Topic And Main Contributions:**

This paper investigates whether a visually grounded PCFG (vc-PCFG) model trained on a source domain of image captioning data can be transferred to other text domains. The original vc-PCFG model does not use pretrained word embeddings like Glove, making it hard to transfer across domains due to vocabulary mismatch. Therefore, this paper proposed tvc-PCFG by replacing the word embeddings in the original model with Glove embeddings that are kept frozen during training, and using domain-specific vocabulary at test time.

Experiments on several domains including both image captioning data and text-only data show that the proposed method transfers well to similar text domains but struggles on domains that are different from the training domain. The paper also conducts extensive ablation studies and analysis to show that domain similarity is indeed correlated with transfer performance, and that lexicon coverage is the most important factor in determining transferability.

Main contributions:

a) Empirical results showing that vc-PCFG models can achieve good cross-domain performance when using pretrained embeddings.

b) Detailed error analysis showing that lexical discrepancy is the main challenge in transferability of vc-PCFG models.


**Questions For The Authors:**

a) I don’t quite understand the PERM baseline, why would its performance indicate whether the model exploits lexical information or not?

b) In Table 1, do you have a hypothesis on why using pre-trained Glove leads to much worse performance when the model is only trained on text data of MSCOCO? This seems counter-intuitive.

c) In Table 2, why adding visual grounding to the in-domain Flick30k model hurts performance, while adding images from the MSCOCO dataset greatly improves performance in the transfer-learning setting?

d) In the L10 variants, why don’t we set the length cutoff to a large number so that the average length of the subsets is closer to that of the training set? It might make it easier to compare the results.


**Reasons To Accept:**

a) This paper studies an interesting problem of cross-domain transfer of visually grounded PCFG models.

b) Very comprehensive experiments on various potential factors for transferability and careful analysis.


**Reasons To Reject:**

a) This paper focuses on one single aspect of one specific model, transferability of visually grounded PCFGs, which might not be of general interest to the field.

b) This paper is a bit hard to follow, especially for readers not familiar with the original vc-PCFG work. I’ll list my questions and comments below.


**Reproducibility:**

4: Could mostly reproduce the results, but there may be some variation because of sample variance or minor variations in their interpretation of the protocol or method.

**Reviewer Confidence:**

3: Pretty sure, but there's a chance I missed something. Although I have a good feel for this area in general, I did not carefully check the paper's details, e.g., the math, experimental design, or novelty.

**Typos Grammar Style And Presentation Improvements:**

a) L252, the term “c-PCFG” is used without definition (not until page 6).

b) L288, the performance improvement should be 20.8% instead of 10.8%?

---

> ### Author Rebuttal · Authors · 2023-08-28
>
> Thank you for providing a detailed review and insightful comments. We would like to clarify the contribution and novelty:
>
> - **A novel research question:** To the best of our knowledge, we are the first to explore transferability in grammar induction. Our aim is to discern whether a parser truly induces a grammar of the language (i.e., English), or if it merely learns rules or shortcuts specific to a particular domain. This exploration is especially important within the context of grounded learning. Given that groundings can only be obtained for specific domains and certain language expressions, the transferability of the acquired syntactic expertise becomes crucial.
>
> - **A novel model:** Apart from conducting a variety of evaluations and analyses, we present a simple yet effective transfer learning model. Our model also provides an inspiration for developing multilingual transfer learning models for grammar induction as discussed in line 499.
>
> Q: *relying on vc-PCFG*
>
> We agree that our work relies on technical details of vc\~PCFG and will expand on vc\~PCFG in Section 2.1 to make our paper self-contained.
>
> Q: *the PERM baseline*
>
> In PERM baseline, we randomly exchange the inferred trees for sentences of the same length. After the exchange, a tree may not correspond to the associated sentence, i.e., the tree can be seen as being inferred without using the correct lexical information. Our hypothesis would be proven to be true if the PERM baseline has a similar performance to tvc~PCFG.
>
> We will revise the paper to make it clear.
>
> Q: *tc\~PCFG underperforms c~PCFG on MSCOCO*
>
> We speculate that domain-specific lexical information is important for grammar induction models. GloVe has been pre-trained on diverse text and may not best reflect lexical information relevant to the domain of MSCOCO captions (e.g., wrong senses and parts of speech), so tc\~PCFG underperforms c~PCFG.
>
> But visual groundings are specific to a domain and could regularize a parser to capture domain-specific lexical information [1], so tvc\~PCFG is less prone to the same issue as in tc\~PCFG; instead, it might be making the best of both visual groundings and pre-trained GloVe, so it outperforms vc~PCFG.
>
> Q: *why adding visual groundings hurts model performance on Flickr30k but is helpful on MSCOCO*
>
> We do not have a good explanation for this but also the variances across runs with (v)c\~PCFG are high.
>
> Q: *better L10 variants*
>
> We agree that a better length cutoff is the one that makes the average length similar to that of the MSCOCO training set, but this method has its own issues: it may incorporate sentence lengths that are not covered by MSCOCO captions, given that MSCOCO captions are generally short; accordingly, further comparison will not be completely fair, either.
>
> Moreover, we use the L10 variant to show that training on the target domain is helpful. Though our L10 variant has a smaller average sentence length, it outperforms tvc\~PCFG, so it is adequate for supporting our hypothesis. But again, we agree with you that using the better length cutoff would make our experiments more comprehensive.
>
> [1] Yanpeng Zhao and Ivan Titov. Visually grounded compound PCFGs. In EMNLP 2020.

---

### Official Review · Reviewer_jC7f · 2023-08-07

**Soundness:** 4

**Excitement:**

3: Ambivalent: It has merits (e.g., it reports state-of-the-art results, the idea is nice), but there are key weaknesses (e.g., it describes incremental work), and it can significantly benefit from another round of revision. However, I won't object to accepting it if my co-reviewers champion it.

**Paper Topic And Main Contributions:**

The paper suggests a way of doing domain transfer of visually-grounded PCFGs by using pretrained GloVe vocabulary embeddings to get the terminal rules. This way, the trained model is not specific to a limited vocabulary, and can use the embedding space to get semantic components of unseen words in new domains. The results show that visual grounding does help in the transfer scenario (compared to no visual grounding) but that transfer is not that good for remote domains.

**Questions For The Authors:**

(A) Why do vocabulary cutting and unks (line 236) when you’re using the GloVe embeddings? Wouldn’t it be simpler to just restrict the vocabulary to all GloVe words?

(B) Line 280 you say that using pretrained GloVe leads to a reduction in performance — isn’t it also the case that tc~PCFG is transfer learning while c~PCFG is tested in-domain?

(D) Line 511 is it possible to use gradient accumulation to make the batch size artificially bigger (sorry, I’m not familiar with the mechanics of PCFG training and this might be a bad question!)

**Reasons To Accept:**

This paper has a solid set of experiments with good baselines that cover many datasets, and further experiments in Data Analysis and Error Analysis that dive into trying to understand some of the initial results. I also think that this paper introduced a neat idea to PCFG parsing: the idea of using pretrained embeddings to not have to worry about vocabulary transfer and be able to check the syntactic viability of transfer.

**Reasons To Reject:**

No clear reasons to reject.

**Reproducibility:**

4: Could mostly reproduce the results, but there may be some variation because of sample variance or minor variations in their interpretation of the protocol or method.

**Reviewer Confidence:**

3: Pretty sure, but there's a chance I missed something. Although I have a good feel for this area in general, I did not carefully check the paper's details, e.g., the math, experimental design, or novelty.

**Typos Grammar Style And Presentation Improvements:**

Table 3: Is the 3rd and 4th row (the c~PCFG) from training on WSJ? I would perhaps make this clearer

Line 259: The PERM baseline and the discussion around lexicalization might merit its own subection or paragraph, it’s a bit odd under Test-time vocabulary. I would make it a separate paragraph heading starting at line 254

It might be better to introduce what S-F1 and C-F1 mean a bit earlier, and they could maybe also be the first columns of the table rather than the last.

Figure 5: there’s a lot of information on this figure, it might be nice to make it a bit sparser. Do all the points/bars use the same y-axis?

Some minor suggestions and typos:

 Line 52: It’s unclear where the T comes from and why it stands for pretrained word embeddings. It might be clearer to make that explicit.

Line 87: It might be nice to have a bit of the explanation before the function. As it stands the reader sees a function where every variable is unknown and then has to read the following paragraph to understand

Line 168: double parentheses, could use \citet instead

---

> ### Author Rebuttal · Authors · 2023-08-28
>
> Thank you very much for your careful reading and insightful suggestions.
>
> Q: *Why do vocabulary cutting and unks  (line 236) …?*
>
> We cut vocabulary for fair comparison (explained in line 251). Note that c\~PCFG uses a vocabulary cutoff, i.e., the top 10,000 frequent words in the training set. Such a strategy is used mainly for better training/test efficiency. To make a fair comparison with c\~PCFG, we use the same vocabulary for t(v)c\~PCFG at test time rather than using the MSCOCO vocabulary, but our embeddings come from GloVe.
>
> Q: *Line 280 … isn’t it also the case that tc\~PCFG is transfer learning while c\~PCFG is tested in-domain?*
>
> Yes. tc\~PCFG is a transfer learning model and c\~PCFG is the original model. The difference is that tc~PCFG uses GloVe.
>
> Line 280 discusses model performance on MSCOCO. Both tc\~PCFG and c\~PCFG are trained on MSCOCO without using visual groundings and are tested on the MSCOCO test set. Thus, this is actually not an evaluation of transferability; instead, we wanted to see if GloVe improves c~PCFG.
>
> Regarding why GloVe hurts the performance of c\~PCFG. We speculate that domain-specific lexical information is important for grammar induction models. GloVe has been pre-trained on diverse text and may not best reflect lexical information relevant to the domain of MSCOCO captions (e.g., wrong senses and parts of speech), so tc\~PCFG underperforms c~PCFG.
>
> Q: *Line 511 … use gradient accumulation to make the batch size artificially bigger…*
>
> We actually tried gradient accumulation. While a larger batch size makes training more stable, it does not bring significant performance improvements and instead makes learning inefficient, e.g., we will have to always keep a copy of gradients during training. Thus, we kept using a batch size of 5.
>
> Thank you again for your suggestions on presentation. We will incorporate them in the final version.

---

### Official Review · Reviewer_ZVfX · 2023-08-11

**Soundness:** 4

**Excitement:**

3: Ambivalent: It has merits (e.g., it reports state-of-the-art results, the idea is nice), but there are key weaknesses (e.g., it describes incremental work), and it can significantly benefit from another round of revision. However, I won't object to accepting it if my co-reviewers champion it.

**Paper Topic And Main Contributions:**

This paper conducts a thorough empirical evaluation of the transferability of visually grounded PCFGs. To address the mismatching vocabulary issue in the transferred domain, pretrained word embeddings are used in the model’s input layer. Evaluations are performed on both proximate-domain (Flickr30k) dataset, and remote-domain datasets (WSJ, Brown, and Enweb), which suggest the benefits from using visual grounding transfer to similar domains but fail to transfer to remote domains. Further analyses reveal that, comparing to other factors like distribution of phrasal labels and grammar rules, the distribution of words/lexicons has more impact in the transferability.

**Questions For The Authors:**

- Table 3, 4, 5: why isn't vc-PCFG listed as a baseline in the remote domains?
- Line 237: given that WSJ, Brown, and Enweb covers more topics than MSCOCO and Flickr, they might also need larger vocabularies. Is limiting the vocabulary size to 10k fair for these datasets?
- Line 279: it’s interesting to see tvc-PCFG outperforms vc-PCFG while tc-PCFG underperforms c-PCFG. Any explanation?
- Table 4: it would be good to have an explanation for each domain abbreviation in Brown dataset.
- Line 336: it’s claimed that longer average length of sentences used in training improves performance. However, there are two varying factors in the comparison: average length and dataset. It’s not accurate to attribute the improvements totally to the longer sentence lengths.
- Line 466: is it possible to measure how statistically reliable the observation is?

**Reasons To Accept:**

The evaluations conducted in the paper have good coverage and a lot of details. One proximate-domain dataset and three remote-domain datasets are included.

**Reasons To Reject:**

The main conclusion from this paper, transferability on similar domains is better than remote domains, has been widely observed in other transfer learning tasks. The further analysis suggests the lexicon overlap of the new domain is the most important factor in the transferability, which is consistent with the findings in c-PCFG in the previous work of Zhao & Titov 2021. The findings in the paper are definitely valuable, but, to some extent, are also a bit incremental to me.


**Reproducibility:**

3: Could reproduce the results with some difficulty. The settings of parameters are underspecified or subjectively determined; the training/evaluation data are not widely available.

**Reviewer Confidence:**

2: Willing to defend my evaluation, but it is fairly likely that I missed some details, didn't understand some central points, or can't be sure about the novelty of the work.

---

> ### Author Rebuttal · Authors · 2023-08-28
>
> Thank you for providing a detailed review and insightful comments. We would like to clarify the contribution and novelty:
>
> - **A novel research question:** To the best of our knowledge, we are the first to explore transferability in grammar induction. Our aim is to discern whether a parser truly induces a grammar of the language (i.e., English), or if it merely learns rules or shortcuts specific to a particular domain. This exploration is especially important within the context of grounded learning. Given that groundings can only be obtained for specific domains and certain language expressions, the transferability of the acquired syntactic expertise becomes crucial.
>
> - **A novel model:** Apart from conducting a variety of evaluations and analyses, we present a simple yet effective transfer learning model. Our model also provides an inspiration for developing multilingual transfer learning models for grammar induction as discussed in line 499.
>
> - **Comparison with Zhao & Titov 2021:** Zhao & Titov 2021 only perform testing in-domain. While they do observe new/unseen words present a challenge, they do not show that it is the main challenge preventing generalization to other domains. We systematically study different types of divergencies and quantify their contribution to the drop in the performance.
>
> Q: *Table 3, 4, 5: why isn't vc~PCFG listed as a baseline in the remote domains?*
>
> WSJ, Brown, and Enweb have no gold visual groundings, so we could not train vc~PCFG on them.
>
> Q: *Line 237: ... WSJ, Brown, and Enweb … Is limiting the vocabulary size to 10k fair for these datasets?*
>
> The reason for keeping the same vocabulary cutoff of 10k is to make fair comparisons across data sets, i.e., eliminating the vocabulary size variable. But, if the goal is to optimize model performance on each data set, the vocabulary size can be treated as a hyperparameter tuned for each data set individually.
>
> Q: *Line 279: … tvc-PCFG outperforms vc-PCFG while tc-PCFG underperforms c-PCFG.*
>
> We speculate that domain-specific lexical information is important for grammar induction models. GloVe has been pre-trained on diverse text and may not best reflect lexical information relevant to the domain of MSCOCO captions (e.g., wrong senses and parts of speech), so tc\~PCFG underperforms c~PCFG.
>
> But visual groundings are specific to a domain and could regularize a parser to capture domain-specific lexical information [1], so tvc\~PCFG is less prone to the same issue as in tc\~PCFG; instead, it might be making the best of both visual groundings and pre-trained GloVe, so it outperforms vc~PCFG.
>
> Q: *Table 4: … to have an explanation for each domain abbreviation in Brown dataset.*
>
> We described them in Table 6 but will definitely elaborate on them in the data set section in the final version.
>
> Q: *Line 336: … not accurate to attribute the improvements totally to the longer sentence lengths.*
>
> Thank you for pointing it out. We agree and will rephrase the statement.
>
> Q: *Line 466: is it possible to measure how statistically reliable the observation is?*
>
> We observe that the ratio decreases as the sentence length increases, i.e., the chance of making mistakes increases.
>
> To measure correlation between the ratio and the sentence length, we compute the Spearman's Rank correlation coefficient for a particular unknown token number (see Figure 6). We chose unknown token numbers 0, 1, and 2 because they are covered by all sentence buckets. Here are the results:
>
> | Number of unknown tokens    | Coefficient | P-value |
> | -------- | ------- | ------- |
> | 0 | -0.917    | 0.0005    |
> | 1 | -0.833      | 0.005    |
> | 2 |-0.983     | 0    |
>
>
> Thus, the results confirm that our observation is statistically reliable.
>
> [1] Yanpeng Zhao and Ivan Titov. Visually grounded compound PCFGs. In EMNLP 2020.

---

### Meta-Review · Area_Chair_ALBt · 2023-09-19

**Recommendation:** 4

**Metareview:**

This paper conducts a thorough empirical evaluation of the transferability of visually grounded PCFGs. The experiments are very solid, covering several datasets with detailed analysis. The introduced idea is also intersting. The major drawback is that as an emperical study, the findings in this paper are plain, which makes the contributions of this paper incremental.

---

### Decision · Program_Chairs · 2023-10-07

**Decision:**

Accept-Findings

**Comment:**

This paper conducts a thorough empirical evaluation of the transferability of visually grounded PCFGs. The experiments are very solid, covering several datasets with detailed analysis. The introduced idea is also intersting. The major drawback is that as an emperical study, the findings in this paper are plain, which makes the contributions of this paper incremental.